# New PEDOT Derivatives Electrocoated on Silicon Nanowires Protected with ALD Nanometric Alumina for Ultrastable Microsupercapacitors

**DOI:** 10.3390/ma15175997

**Published:** 2022-08-30

**Authors:** Marc Dietrich, Loïc Paillardet, Anthony Valero, Mathieu Deschanels, Philippe Azaïs, Pascal Gentile, Saïd Sadki

**Affiliations:** 1CEA, Grenoble INP, CNRS IRIG-SyMMES UMR 5819, University Grenoble Alpes, F-38000 Grenoble, France; 2CEA, Grenoble INP, IRIG-Pheliqs, University Grenoble Alpes, F-38000 Grenoble, France; 3CEA-LITEN-DEHT, University Grenoble Alpes, F-38000 Grenoble, France

**Keywords:** pseudo-supercapacitors, electroactive conducting polymers, SiNWs, 3D nanostructured electrodes, ALD, SDS and SDBS surfactants, PEDOT derivatives

## Abstract

This work deals with electroactive conducting polymers (ECPs) used as a complementary component on purely capacitive silicon nanowires protected by a 3 nm alumina layer. Accordingly, in this work, we use a fast and simple deposition method to create a pseudocapacitive material based on the electropolymerization in aqueous micellar media (SDS and SDBS 0.01 M) of hydroxymethyl-EDOT (EDOT-OH) onto 3 nm alumina-coated silicon nanowires (Al_3_@SiNWs). The composite material displays remarkable capacitive behavior with a specific capacitance of 4.75 mF·cm^−2^ at a current density of 19 µA·cm^−2^ in aqueous Na_2_SO_4_ electrolyte.

## 1. Introduction

On-board electrochemical energy storage is critical to meet the increasing demand for low-power portable devices such as micro-electro-mechanical systems (MEMs), autonomous sensor networks, radio frequency identification (RFID), or biomedical implants [1,2]. Miniaturized electrochemical supercapacitors that can be integrated on circuit chips will be essential for the development of future microelectronic devices due to their ability to provide high power density within an extended lifetime and wide operating temperature range. However, state-of-the-art microsupercapacitors (µSCs) (based on carbon nanotubes, onion-like, carbide-derived carbons) still suffer from a limited energy density, facing comparison with recent Li-ion microbatteries [3,4] Although commercially available supercapacitors can deliver much higher energy density (~5 Wh kg^−^^1^) than traditional solid-state electrolytic capacitors, it is still significantly lower than that of batteries (up to 200 Wh kg^−^^1^). In order to close the energy density gap, hybrid electrodes combining the properties of a high surface area nanostructured support and the large pseudocapacitance of faradic materials holds great promise. During the last decade, the inherent properties of electroactive conducting polymers (ECPs) have also attracted a great deal of attention in the field of electrochemical energy storage devices, comprising mainly battery and supercapacitor devices. ECPs are playing a key role in the category of pseudosupercapacitors due to their energy storage mechanisms associated to faradaic reactions, delivering higher capacitance values compared to pure EDLCs [5].

Within this context, ECPs have demonstrated an enormous potential in terms of high specific capacitance (values ranging from 300 to 800 F·g^−1^ [6,7,8] depending on electrochemical performance conditions), high conductivity (up to 103 S·cm^−1^) [7,9], light weight and flexibility [10], relatively fast charge-discharge processes [11], easy processing and relative low-cost [12,13], and environmental friendliness [14]. Thus, from the synthesis perspective, the electrochemical deposition of ECPs by means of electrochemical techniques based on galvanostatic [15,16,17], potentiostatic [18,19], and potentiodynamic [20,21] methods, recently opened up new perspectives to design and develop a great variety of polymer morphologies. Yet, pseudo-capacitive materials (either transition metal oxides or conjugated polymers) only display a reduced electroactive window and therefore lead to low operating voltage not exceeding 2.0 V [22]. As the specific energy of a supercapacitor is proportional to the square of the cell voltage, raising the energy density will ultimately require the use of asymmetric electrodes with complementary potential ranges [23,24] which is reachable here through the design of n-doped and p-doped associated ECPs as different electrodes materials.

The objective of this work is to increase the areal energy density of µSCs. Integrated micro-devices will be built with nanocomposite electrodes based on silicon nanowires (SiNWs) as the conductive nanostructured substrate [25,26] and ECPs as the key energy source for the electrode. Such nanowires were used as electrode materials and gave double layer capacitance values reaching 46 µF·cm^−2^ for hundreds of cycles [27]. Silicon nanostructures are prepared thanks to the chemical vapor deposition (CVD) growth process (bottom-up) with doping precursors to keep a high conductive material as current collector [28,29,30,31]. Specific ECPs are then electrodeposited in aqueous media [32] as they are designed to have improved chemical compatibility with the polar alumina layer at the surface of the SiNWs. To our knowledge, this project would represent the first time that ECPs are coated on Al_3_@SiNWs as an electrode material for micro-supercapacitor application. Those ECPs will be electropolymerized directly on the Al_3_@SiNWs with the help of an aqueous solution where the concentration of surfactant is high enough to reach critical micellar state, in presence of sodium dodecyl sulfate (SDS 0.01 M) and sodium dodecyl benzene sulfonate (SDBS 0.01 M) as different surfactants compared in this study [33,34,35,36]. In this work, we study the ability to deposit ECPs through critical micellar media and compare the salt effect on the electrochemical deposition mechanism of EDOT and EDOT-OH on passivated SiNWs as an active electrode material. Both nanocomposite electrodes are electrochemically characterized to show the new increase of energy density.

## 2. Materials and Methods

### 2.1. SiNWs Electrodes

Highly n-doped silicon wafers (100 mm diameter, ⟨111⟩ orientation, Ω < 0.005 ohm·cm) from Silicon Materials Inc. (Glenshaw, PA, USA) were used as the base material for the growth of SiNWs. Prior to any surface modification, native SiO_2_ from passive oxidation was removed by etching the samples in hydrofluoric acid with a HF Vapor phase etcher process (Primaxx Monarch3 from SPTS), 1 min gas exposure for slow native oxides, followed by a nitrogen drying. Wafers were then coated with 4 nm gold thin film by metal evaporation (Evaporator MEB550 from PLASSYS). Samples were then diced into 10 × 10 mm^2^ squares and used as substrates for SiNWs growth after extensive rinsing in acetone, isopropanol.

Silicon nanowires were grown by CVD using the vapor liquid solid method (VLS). Previously deposited gold film was used as the VLS catalyst for the growth. The nanostructure growth was conducted in a hot wall quartz tube CVD reactor, at 650 °C and under 6 Torr total pressure, using 700 sccm H_2_, 40 sccm SiH_4_, and 90 sccm PH_3_ (0.2% PH_3_ in H_2_), respectively, as carrier, silicon precursor, and doping gases. An extra 100 sccm HCl gas was added during the growth in order to reduce surface gold migration and enhance the morphologies of the SiNWs and the doping as previously reported [37,38]. The growth rate under these conditions was about 500 nm·min^−1^ and the fabricated SiNWs were tuned to be 50 μm long. For the catalyst deposited from metal evaporation, the mean SiNWs diameter was estimated around 61 nm ± 26 nm from SEM measurements. SiNWs density was roughly estimated to be 7 × 10^7^ SiNWs cm^−2^.

### 2.2. ALD of Alumina on SiNWs

Nanometric atomic layer deposition of thermal Al_2_O_3_ thin films was carried out using trimethylaluminum (TMA) and H_2_O as precursors in a Fiji200 reactor (Ultratech, Mumbai, India). The previously HF-deoxidized samples were placed in the deposition chamber, under 250 °C, 10^−2^ Torr, argon purge gas, and an automated recipe alternating precursors injection steps and purge steps (0.06 s triethylaluminium (TMA), 8 s purge, 0.06 s H_2_O, and 8 s purge) was performed until the desired number of cycles was reached. The deposition rate was estimated to be 0.92 Å·cycle^−1^ (thus a 3 nm thick layer requires 33 ALD cycles). SiNWs samples were referred to as Al_3_@SiNWs for a 3 nm Al_2_O_3_ homogeneous and conform cover.

### 2.3. Electropolymerization of EDOT and EDOT-OH

Electropolymerization was performed in a three-electrode cell, using the Al_3_@SiNWs as the working electrode for polymer electrodeposition, with platinum coil counter electrode and an Ag/AgCl reference electrode. The electrolyte containing the EDOT and EDOT-OH (8 mM) monomers was a critical micellar solution with either SDS (0.01 M) or SDBS (0.01 M) as surfactant. After 1 h stirring, a 1 mL volume of solution was injected in the cell and cycled at 50 mV·s^−1^ from −1.1 to 1.1 V vs. Ag/AgCl. The quantity of polymer electrodeposited was observed in the first time after 20 cycles for experimental checking and after 100 cycles for electrochemical characterizations, showing the Al_3_@SiNWs turning dark blue. Samples were examined through scanning electronic microscopy (SEM) (Zeiss Ultra 55, Zeiss, Oberkochen, Germany) and energy dispersive X-ray spectroscopy (EDX) to confirm the polymer deposit and morphology.

### 2.4. Electrochemical Measurements

Electrochemical measurements were carried out in an aqueous electrolyte of Na_2_SO_4_ 0.7 M electrolyte at room temperature. The electrolyte was initially purified from oxygen using a 10 min argon bubbling then injected inside each cell with a 1 mL syringe under the consistent basis of roughly 300 μL for each test. The electrochemical characterization tests were conducted with a potientiostat/galvanostat (VMP3, Biologic, France). The investigation of the nanocomposite µSCs electrodes was performed in a commercial electrochemical cell (ECC cell, EL-CELL, Hamburg, Germany) in both three electrodes and two electrodes cell configuration. Three electrodes cell characterization was conducted using the silicon-based sample (active area roughly 0.4 cm²) as the working electrode, a platinum coiled wire as counter electrode, and an Ag/AgCl reference electrode for aqueous measurements. Preceding any experiment, the silver pseudo-reference electrode was calibrated using the voltammetric response of ferrocene redox couples in Na_2_SO_4_ 0.2 M using an Ag/AgCl reference electrode. Two electrodes cell devices were assembled by sandwiching a 0.7 M Na_2_SO_4_ soaked Whatman grade 41 paper separator between two identical electrodes for a symmetrical system. Cyclic voltammetry (CV) was conducted with various potential windows ranging from −0.9 to 0.9 V vs Ag/AgCl reference electrode at different scan rates, from 50 mV·s^−1^ to 1 V·s^−1^. The first cycles were considered as stabilization state cycles and only following cycles are displayed here.

Galvanostatic measurements (GCPL) of symmetric systems (view Appendix A for detail) were carried to up to 1.2 V at different charging rates. Charge-discharge measurements were conducted at multiple current densities (ranging from 0.05 mA·cm^−2^ to 1 mA·cm^−2^). Those were used to obtain areal capacitance, energy, and power densities derived from the slope of the galvanostatic measurement, taking into account the footprint of the µSC device (1 cm^−2^). Capacitance retention of the supercapacitor configuration was investigated through long-term charge-discharge of symmetric devices in Na_2_SO_4_ 0.7 M tests at a current density of 0.5 mA·cm^−2^.
$$C=\frac{I\Delta t}{\Delta U\;\times \;S}$$

For GCPL (1), where SC (F·cm^−2^) is the specific areal capacitance of the microsupercapacitors, I (A) corresponds to the discharge current, ∆U (V) is the potential change over the range applied and S (cm) the surface of the device tested.
$$C=\frac{\int_{U1}^{U2}\mathrm{IdU}}{2\nu \times \Delta U}$$

For CV (2), extraction of the capacitance is used with the following equation, where the CV surface is integrated over dU (V) its full potential range, over ν (V·s^−1^) the scan speed, and ∆U (V) the potential window of the cell. All electrochemical characterizations were performed several times with samples from separate CVD and ALD batches using the same conditions to ensure a good reproducibility in the experimentation and presented data.

## 3. Results and Discussions

### 3.1. Electropolymerizatoin on Al_3_@SiNWs

#### 3.1.1. Electropolymerization of EDOT and EDOT-OH

The electropolymerization step uses nano-structured silicon-based electrodes to make a new nanocomposite for supercapacitor applications. The electrode is made of n-doped silicon nanowires and a thin 3 nm Al_2_O_3_ layer covers them homogeneously. This alumina layer gives new vision on the potential applications of our system: such protected electrodes are still giving good electrochemical performances while being compatible with aqueous media, which is not the case for pristine silicone electrodes. Previous work showed that the electropolymerization of EDOT on SiNWs has already been made using ionic liquids as PYR_13_ TFSI as transport media [17]. However, with this alumina layer, we have the possibility to repeat this procedure within aqueous media, aiming to display the new interfacial properties with the 3 nm Al_2_O_3_ protecting layer, we compared the effect of two monomers: EDOT and EDOT-OH. To enhance the ability to deposit ECPs, and because EDOT monomers are poorly miscible in aqueous media, for both polymers, we use two surfactants (SDS and SDBS) to help depositing locally our pseudocapacitive material. With the surfactant addition and upon vigorous stirring, micelles are able to imprison the EDOT and EDOT-OH in their hydrophobic core. This conformation allows the monomers to move freely in the aqueous media, diffuse through the tortuosity of the SiNWs entanglement, and react at our electrode interface during the process of electrodeposition. SDS and SDBS are here compared regarding their ability to create proper conditions for electrodeposition, with different chemical properties and an increased steric hindrance for SDBS that could impact the micellar formation [39].

For the CV curves of the electropolymerization process (Figure 1a), successive cycles display higher current densities because of the previously electrodeposited polymer getting oxidized, displaying first shapes of pseudocapacitance [40]. It clearly shows that our monomers are getting to their radical EDOT states at the end of the oxidation process around 1.1 V vs. Ag/AgCl, allowing the electropolymerization and deposition on Al_3_@SiNWs, thus nucleating and creating a polymer film on our silicon electrodes. After 20 cycles of electropolymerization, we can assume that the deposition of the EDOT-OH monomer has a better yield than the EDOT monomer with higher current densities reached and a less resistive CV oblique shape. Here, the effect of the hydroxyl function allows a better solubility of the chemicals in aqueous media, as well as a higher affinity with the polar alumina layer. SEM imagery (Figure 1b) shows the top view of the nucleation of the electroactive polymers seems also to be favored at the apex of our Al_3_@SiNWs thanks to the field effect and propagate to the trunks of the wires after the various electropolymerization cycles. It creates larger areas, much thicker with the characteristic cauliflower structure for both PEDOT and PEDOT-OH and it propagates to the length of the Al_3_@SiNWs as a thinner layer.

#### 3.1.2. Electropolymerization Mechanism of EDOT-OH

For both the EDOT and the EDOT-OH monomers, the electropolymerization mechanism is as described in Figure 1c [41,42,43,44]. The monomer reacts at the SiNWs electrode surface with the transferred electrons, creating mesomer forms leading to the positive radical form of the monomer at the reached potential of 1.0 V vs. Ag/AgCl. Thus, upon the reduction part of the CV, the radical forms are going to react together, liberating H_2(g)_ and creating a first dimer. This first assembly of polymer will be able to be re-oxidized in the following oxidation part of the CV, appearing here as a stacking capacitive peak with the appearance of the polaronic forms. Those will be able to react at one of its extremities of opened sites on the thiophene ring, extending the polymer chain and continuing mechanism until the CV is stopped.

#### 3.1.3. Effect of Surfactants SDS and SDBS

Because of the better electrochemical response of the EDOT-OH monomer, the following experiments were focused on that specific monomer. We are comparing the effect of two surfactants (SDS and SDBS) on the quality and efficiency for an electrodeposition of EDOT-OH in critical micellar media [45,46,47]. We can see on the voltammograms (Figure 1d) that SDS is more suited to deposit larger amounts of polymer on the nanostructures. The chemicals reasons are here described as the steric hindrance of the SDBS with the benzene group, limiting the size of the micelle and hindering the diffusion and exchanges at the electrode interface. The SDBS surfactant is showing slower kinetics for the EDOT-OH electropolymerization to reach the polaronic and bi-polaronic states and might be related to different pattern for the deposition mechanism. The traces of PEDOT-OH using SDBS as a surfactant are scarce on the SEM observations, only appearing as small grains scattered across the Al_3_@SiNWs. However, in SDS, the nucleation of the PEDOT-OH is faster and SEM micrographs shows that we are able to deposit polymer on the Al_3_@SiNWs. Supposition was rising that the alumina layer could play the role of a limitation barrier because of its insulator behavior, limiting electron transfer and protecting our SiNWs from electrodeposition of EPCs. However, as a polar material, the alumina improves the chemical binding between Al_3_@SiNWs and the PEDOT-OH.

#### 3.1.4. EDX Analysis

On the PEDOT-OH in SDS samples, EDX scans were probed at two distinct areas of the nanocomposite, the apex of the Al_3_@SiNWs and the length surface at the basis of the wire. The apex (Figure 2a) presents large nucleation areas of electrodeposited polymer, exhibiting high intensity peaks for S, C, and O, matching with the PEDOT-OH chemical composition ratios. Small traces of Na for residue of the surfactant and Al for the protective alumina layer appear too on the spectrum. To check the presence of thin polymer film on the main part of the wire (Figure 2b), EDX probing shows higher ratio for S (6.37%) to C and O (24.44% and 7.84%), fitting with the deposition of a thin layer of polymer added to the alumina layer beneath it. Those results could benefit from TEM vision for a better approach across the PEDOT-OH deposited on the depths of the SiNWs entanglement and the presence of the ECPs thin film. Although, the electrodeposition mainly occurs on the apex of the Al_3_@SiNWs, that could be due to the presence of a highly conductive material in the presence of the gold marble (leftover catalyst from the CVD growth). However, the gold marbles are also covered with alumina during the fabrication process of the electrodes which means that the electrodeposition is possible even with a dielectric layer fully covering the substrate. This confirms the compatibility of nanometric high-k dielectrics with electrochemical applications.

### 3.2. Characterization of PEDOT-OH in SDS and SDBS

#### 3.2.1. Electrochemical Characterization

The CV characterization of our nanocomposite electrode was realized in a three-electrode cell, from −0.9 to 0.9 V vs. Ag/AgCl in a 0.7 M Na_2_SO_4_ electrolyte and comparing the CV results (Figure 3a) of EDOT-OH electropolymerization in SDS, SDBS, and pristine Al_3_@SiNWs electrodes. We can clearly observe the shape of oxidation of EDOT-OH at −0.4 V vs. Ag/AgCl and an increasing current density response until a plateau from −0.3 to 0.8 V vs. Ag/AgCl, leading to improved specific capacitance of our electrodes. Extracting the specific capacitances related to their respective scan rate (Figure 3b) allows us to compare our systems properties to others. The PEDOT-OH electropolymerized in SDS present high specific capacitance with 32.1 mF·cm^−2^ compared to the protected nanowire and the PEDOT-OH electropolymerized in SDBS with 4.8 mF·cm^−2^. By the addition of PEDOT-OH to our system, at 100 mV·s^−1^, we get more than 30 times the standard pristine Al_3_@SiNWs thanks to the pseudocapacitive contribution of the EPCs.

#### 3.2.2. Salt Effect on PEDOT-OH Morphology

In SEM, visually, more PEDOT-OH was deposited on the Al_3_@SiNWs in SDS conditions (Figure 3c), forming large nucleation spots that propagate along the length of the nanowire creating conical structures. PEDOT-OH deposition in SDBS conditions (Figure 3d) seems to form smaller spherical grains. Hardly observable in the SEM principal electrons detector, a change to secondary electron detector scan mode (SE2) showed small polymer particles all along the nanowires (50 to 200 nm of diameter). For the same concentration of monomers polymerized, the efficiency over 100 cycles of electropolymerization was about 10 times higher for a SDS surfactant. Appearing clearly on symmetrical CV, the current density from the pseudo-capacitive material was reaching 200 mV·s^−1^ up to 3.6 mA·cm^−2^ for SDS and 0.39 mA·cm^−2^ for SDBS electropolymerization method. From those increased scan speed CVs, the polymer film had two different responses regarding the surfactant used. Although the surfactant had a major impact on the efficiency of the electropolymerization, it also influenced the morphology of the deposited PEDOT-OH, leading to different CV responses shapes and peaks slight shifts. Here, the scan speed is not directly proportional to the capacitance and is rather being fit with a square root dependency, showing that the pseudo-capacitance behavior is contributing to the energy storage mechanism. By looking in the literature, these performances are slightly higher than the performances of the PEDOT:PSS on protected SiNWs [48,49] (from 6 to 15 mF·cm^−2^). However, these results must be taken with hindsight since the system with PEDOT:PSS has a narrower electrochemical window, and a better stability to the increase in potential. Indeed, the PEDOT-OH has a weak stability over the increase of the scan rate, since the kinetic of pseudocapacitive processes is determined by slow faradic reactions compared to pure capacitive storage.

#### 3.2.3. Scan Rate Effects on PEDOT-OH Coatings

The nature of the systems PEDOT-OH electropolymerized in SDS and in SDBS is described in (Figure 3e,f). Specific capacitances are extracted by the integration of the cycles of the 100-cycle experiment CV responses divided by the potential window. These capacitances are then plotted vs. the V and √V from the Trasatti method [50,51]. From this data treatment, it is possible to assess the nature of the system, either faradic or capacitive. If the specific capacitance is proportional to the scan rate the system is defined as capacitive and if it is proportional to the square root of the scan rate, the system is pseudocapacitive. As we observe the behavior of our two systems with SDS and SDBS, we can see a clear difference between each: for the SDS deposition, with larger aggregates, we have a clear dependency to a pseudocapacitive mechanism with a fitting with √V of the slope. The capacitance attributed to pseudocapacitive mechanism represents 60.5% of the total capacitance and the other 39.5% belongs to EDLC storage (view Appendix A for details). For SDBS electrodeposited PEDOT-OH, the discrimination between capacitive and capacitive also shows a tendency closer to pseudocapacitive mechanism, but with a lower total capacitance. The hypothesis behind these results is that the polymer has lowered kinetic of electropolymerization in SDBS, which results in a system with less pseudocapacitive material than SDS depositions. The contribution of the polymer to the specific capacitance is lower than the case of electropolymerization in SDS, although both EDLC capacitances are enhanced by the addition of this pseudocapacitive contribution.

### 3.3. Electrochemical Cyclability and Ageing

#### 3.3.1. Effect of Current Densities

The PEDOT-OH deposited on Al_3_@SiNWs was also tested in symmetric devices for performance estimation. The GCPL shape shown in Figure 4a is a sharp triangle indicating the main storage mechanism being capacitive rather than faradic and was tested to different current densities from 19 µA·cm^−2^ up to 380 µA·cm^−2^ and at a cell tension of 1.2 V. It appears that, as shown in Figure 4b, our system can load with a small ohmic drop of 0.04 V after a charge in 600 s at 19 µA·cm^−2^. At higher current densities, and up to 380 µA·cm^−2^, this ohmic drop falls to 0.5 Volts. Those GCPL results show a two-electrode cell supercapacitor displaying a capacitance up to 4.75 µF·cm^−2^ in aqueous electrolyte using a pseudocapacitive material.

#### 3.3.2. Electrochemical Cyclability

The two-electrode symmetrical system has also been exposed to a long life cycling at 0.5 A·g^−1^ (for an active mass of 0.62 mg^−1^ of EDOT-OH deposited) current density to up to 100,000 cycles in aqueous electrolyte Na_2_SO_4_ 0.7M. As shown in Figure 4c, there is a quick capacitance decrease in the first 100,000 cycles, from a starting point at 4.75 mF·cm^−2^, it fades up to stabilizing at 2.61 mF·cm^−2^ and the end of the cycling after a 55% loss in capacity retention. This can also be observed with the comparison of the voltammograms (Figure 4d), showing the loss of capacitance between 20th and after 100,000th cycles. Because the capacitive plateau associated to the conductive oxidized polymer remains horizontal, keeping the square shape of the voltammetry, that could be associated to an increased resistivity in the system. The pellet shape located at the apex of the Al_3_@SiNWs could have been removed and insulated from the electrodes, leading to a quick capacity fade in our system.

## 4. Conclusions

In this work, we proposed a fast and simple deposition method to create a new nanocomposite with pseudocapacitive material based on the electropolymerization in aqueous micellar media (SDS and SDBS 0.01 M) of hydroxymethyl-EDOT (EDOT-OH) onto 3 nm alumina-coated silicon nanowires (Al_3_@SiNWs). It forms a homogeneous thin film coverage on the nanostructures, as proven by SEM and EDX characterizations. This coating allows enhanced electrochemical performances for ECPs and energy storage compatibility with higher energy densities while remaining in aqueous media. The pseudocapacitive mechanism for energy storage is the main energy storage mechanism observed in this kind of system, largely outdoing standard EDLCs supercapacitors. The nanocomposite material produced displays of capacitive behavior with a specific capacitance of 4.75 mF·cm^−2^ at a current density of 19 µA·cm^−2^ in aqueous Na_2_SO_4_ electrolyte. The Al_3_@SinWs-EDOT-OH electrodes also shows improved life cyclability despite falling at 55% after 100,000 cycles in aqueous media at 0.5A·g^−1^ of electroactive material. This confirms the compatibility of thin dielectric layers with energy storage mechanisms. While being able to perform in aqueous solvent, the protected SiNWs can be associated with various pseudocapacitive materials through aqueous deposition techniques. Pseudocapacitive materials such as derivatives from EDOT as EDOT-OH show a better affinity and electropolymerization results with this technique.

## Figures and Tables

**Figure 1 materials-15-05997-f001:**
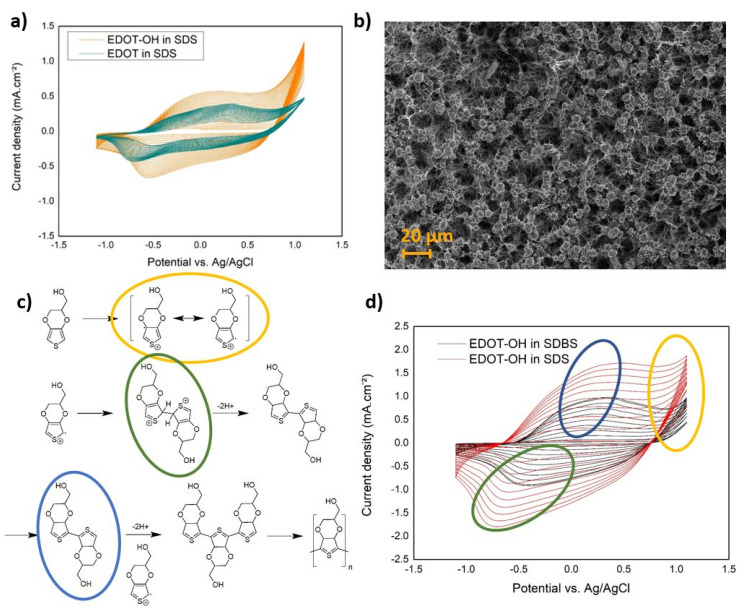
Nanocomposite electrode formulation, with (**a**) the electropolymerization comparison of EDOT and EDOT-OH, (**b**) the SEM top view of the PEDOT-OH-Al_3_@SiNWs nanocomposite electrode, (**c**) electropolymerization mechanism described and main deposition steps specified on the CV curves, with circled in yellow the polaronic form, circled in green the dimerization, circled in blue the propagation of the phenomenon and deposition of polymer EDOT-OH (**d**) comparison of surfactants SDS and SDBS in electrodeposition of EDOT-OH at 50 mV·s^−1^, critical micellar media (0.01 M).

**Figure 2 materials-15-05997-f002:**
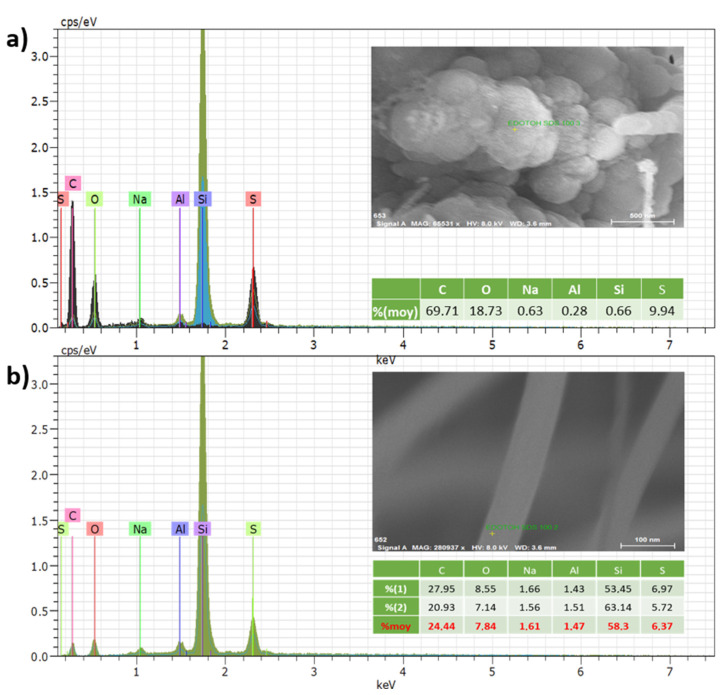
EDX measurements (**a**) of the apex of the Al_3_@SiNWs for large nucleation grains of EDOT-OH in SDS and (**b**) bottom of the Al_3_@SiNWs and traces of EDOT-OH deposited in SDS.

**Figure 3 materials-15-05997-f003:**
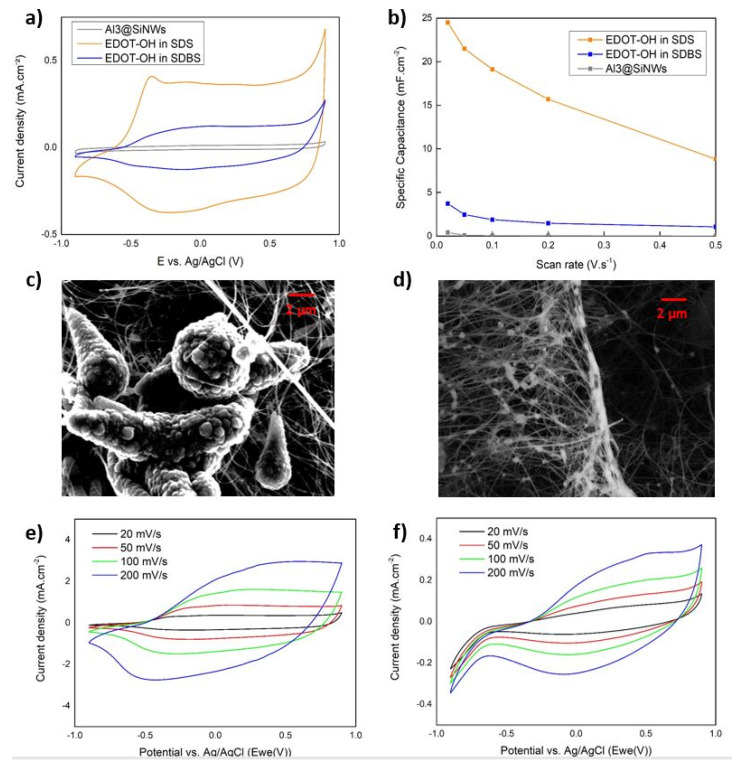
(**a**) Comparison between with the bare Al_3_@SiNWs of the SDS and SDBS 100 cycles electropolymerization in Na_2_SO_4_ 0.7 M and (**b**) their capacitance evolution regarding the scan rate. SEM images of electropolymerization made with (**c**) SDS and (**d**) SDBS in SE2. Cyclic voltammetry at different scanning rate of (**e**) EP in SDS and (**f**) EP in SDBS.

**Figure 4 materials-15-05997-f004:**
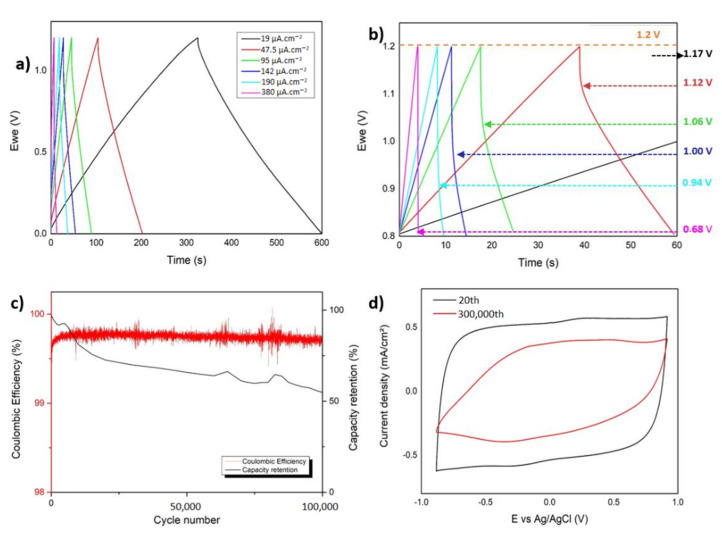
(**a**) Galvanostatic charge/discharge of the symmetric cell of Al_3_@SiNWs-PEDOT-OH electrodes. (**b**) Ohmic drop effect during galvanostatic charge/discharge. (**c**) Electrochemical cyclability of Al_3_@SiNWs-PEDOT-OH in aqueous media and (**d**) CV cyclability comparison between 20th and 100,000th cycle.

## Data Availability

The data underlying this article will be shared on reasonable request to the corresponding author.

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
