# Peer review of "New PEDOT Derivatives Electrocoated on Silicon Nanowires Protected with ALD Nanometric Alumina for Ultrastable Microsupercapacitors"

_materials, 2022, doi:10.3390/ma15175997_

Round 1

Reviewer 1 Report

This work reports the electro-polymerization of EDOT and EDOT-OH on aluminum-coated silicon nanowires by in situ chemical oxidation. The effect of surfactants (SDS &SDBS) in electro-polymerization was described in detail and how it affects the morphological and electrochemical properties of the electrochemically conducting polymers. EDOT-OH polymerized with SDS showed excellent performance in pseudocapacitive energy storage and release in a three-electrode setup. The article presents relevant original results, the text is scientifically correct and justified, and the conclusions are based on the described results. This manuscript may be of interest for publication. However, the authors should consider the following recommendations to significantly improve the quality of the paper:

General comment:

The manuscript must be improved in terms of language. Many sentences are formulated in a colloquial and imprecise manner. In addition, the presentation of the units and axis labels of the figures is not consistent and partly incomprehensible! I ask the authors to revise thoroughly.

(1) Page 1, Line 11-16: The Abstract is very general and more like a short introduction. Please shorten the first two sentences. Also include your observations on the comparison of SDS and SDBS during electro-polymerization and the resulting electrochemical and structural properties in short form.

(2) Page 1, line 32-34: Please proved average values for energy density and capacitance for the mentioned mSCs. What is the benchmark at the moment, where lies the reachable goal for this mSC’s?

(3) Page 1, line 41: The reference is missing.

(4) Page 2, line 57-58: Why were the materials mentioned chosen, why not others? Please make a statement about why the materials are beneficial and why it makes sense to investigate them.

(5) Page 2, line 63-64: The sentence is incomplete or incomprehensible.

(6) Page 2, line 65: “critical micellar media”… what do you mean with critical?

(7) Page 2, line 74: “highly” n-doped silicon wafers - What was the share of endowment, what is " highly endowed "?

(8) Page 5, figure 1: In Figure 1 a and d, the current is normalized to the electrode area. Which area was used for the calculation? For all CV’s the electroactive area needs to be determined, since the aluminum-coated area can have an insulating effect. The same applies to all other cyclic voltammograms.

(9) Page 5, figure 1c: The thiophenyl ring is a bit distorted for most molecules. In addition, the graphic is blurred.

(10) Page 7, figure 3 c and d: Please provide SEM images with more brightness.

(11) Page 7, figure 3f: One can see the exponential increase due to HER. Please provide an answer as to why this is the case with SDBS and not SDS? It might also be better to use a smaller potential window for f) since the electrolyte might have changed during the cycling. Please provide voltammograms of f) with a smaller (less low) potential window to check if this affected the CV.

(12) Page 8, line 270: Please provide capacity retention data for the different scan rates and explain the changes in detail.

(13) Page 8, line 275: The authors compare the performance of their sample with one other. Please provide a more detailed comparison that considers other EPCs with similar morphology or composition. What is the advantage of your samples compared to other samples known in the literature?

(14) Page 8, line 287-290: Please provide respective figures.

(15) Page 9, line 320-322: If possible, can the authors provide additional data for analysis of the aged electrodes, e.g. SEM images…?

(16) Page 9, figure 4c: Labeling of the graph is in French. Please change to English.

Author Response

Dear Colleague

 We appreciate your useful comments about our paper and the careful attention you gave to the scientific details in our manuscript. Your remarks are truly funded and those specific details needed further explanations to be as clear as we would have wanted. That is why we tried our best, in the due time, to respond to all of your requests with efforts and precision: details about language and coherence of the writing were identified and fixed to correspond with the proper phrasing, while some experimental parts also needed to be cleared-out and re-worked to give the proper feeling when reading
our work.

That’s took a re-composing of the graphs and SEM imagery as well as extra data added as a supplementary for the paper, to improve our arguments on our scientific work. In the name of the authors of this manuscript, we want to thank you for your kind interest and your time as well as the professionalism you involved to review this paper.

Best Regards,
Saïd Sadki

Reviewer 2 Report

In their submission to Materials entitled "New PEDOT Derivatives Electrocoated on Silicon Nanowires Protected with ALD Nanometric Alumina for Ultrastable Microsupercapacitors", Sadki and coworkers describe a methodology to prepare composite materials based on the electropolymerization in an aqueous micellar media of EDOT-OH onto alumina-coated silicon nanowires. The composite material has a specific capacitance of 4.75 mF/cm2. The work is well presented and the results of interest. My only concern is that XPS characteriztion is missing; thus I recommend to include a XPS analysis of the prepared samples. Thus, I recomment to publish this paper  after adressing ths concern.

Minor concerns: page 1, line 41: The reference is missing

Author Response

(The authors gave the same response as above.)

Reviewer 3 Report

This paper reports the electropolymerization of hydroxymethyl-EDOT (EDOT-OH) onto 3 nm alumina-coated silicon nanowires (Al3@SiNWs) in aqueous micellar media (SDS and SDBS 0.01 M) for the preparation of a nano-composite for microsupercapacitors. The prepared composite has a moderate performance as a supercapacitor electrode. The prepared composites are not fully characterized. The manuscript must thus undergo extensive revisions before it can be accepted in Materials.

1.               Several grammar and editing mistakes can be found throughout the manuscript. Please try to avoid unnecessary capitalization of random words in a sentence.

2.               Abbreviations that appear for the first time in the article, such as RFID (Page 1 Line 28), ILs (Page 4 Line 153), CV (Page 4 Line 165), EP (Page 4 Line 177), EPCs (Page 5 Line 208), should be given their full name.

3.               Missing reference on page 1 Line 41.

4.               Page 2 Line 58, what is the role of silicon nanotrees (SiNTrs) in the prepared composite materials?

5.               It is suggested to write the materials and methods, results and discussion sections in the past tense.

6.               The equations used to calculate capacitance must be included in the experimental section of the manuscript.

7.               Characterization techniques such as FTIR, XRD etc. should be provided to verify the prepared composites.

8.               A control sample without an alumina layer should be prepared, and its performance should be compared to that of the other samples.

9.               What is the mass loading of PEDOT-OH in PEDOT-OH-Al3@SiNWs nanocomposite?

10.            Page 6 Line 246, The PEDOT-OH electropolymerized in SDBS present high specific capacitance compared to the protected nanowire and the PEDOT-OH electropolymerized in SDBS. SDS is written as SDBS.

11.            Page 8 Line 284, capacitances vs. the V and √V graphs should be provided to assess the nature of the system.

12.            Provide the photographs of the prepared microsupercapacitors.

13.            Electrochemical impedance spectroscopy (EIS) measurements of the prepared samples should be included in the manuscript.

14.            The Y axis of Figure 4.a and b should be written in full form.  In Figure 4c, cycle nbr should be changed to cycle number on the X axis. Figure legends of Figure 4c should be written in English.

Author Response

(The authors gave the same response as above.)

Round 2

Reviewer 1 Report

The authors have improved all the points raised. 

Therefore, I recommend this work for publication!

Reviewer 3 Report

All of the reviewer's concerns have been properly addressed by the authors. So the manuscript can be accepted for publication in the Journal Materials.